# Three New Species of *Microdochium* (*Sordariomycetes*, *Amphisphaeriales*) on *Miscanthus sinensis* and *Phragmites australis* from Hainan, China

**DOI:** 10.3390/jof8060577

**Published:** 2022-05-27

**Authors:** Shubin Liu, Xiaoyong Liu, Zhaoxue Zhang, Jiwen Xia, Xiuguo Zhang, Zhe Meng

**Affiliations:** 1College of Life Sciences, Shandong Normal University, Jinan 250358, China; shubinliu2022@126.com (S.L.); liuxiaoyong@im.ac.cn (X.L.); zhxg@sdau.edu.cn (X.Z.); 2Shandong Provincial Key Laboratory for Biology of Vegetable Diseases and Insect Pests, College of Plant Protection, Shandong Agricultural University, Taian 271018, China; zhangzhaoxue2022@126.com (Z.Z.); xiajiwen163@163.com (J.X.)

**Keywords:** *Ascomycota*, *Amphisphaeriaceae*, taxonomy, multigene phylogeny, new taxon

## Abstract

Species in *Microdochium*, potential agents of biocontrol, have often been reported as plant pathogens, occasionally as endophytes and fungicolous fungi. Combining multiple molecular markers (ITS rDNA, LSU rDNA, TUB2 and RPB2) with morphological characteristics, this study proposes three new species in the genus *Microdochium* represented by seven strains from the plant hosts *Miscanthus sinensis* and *Phragmites australis* in Hainan Island, China. These three species, *Microdochium miscanthi* sp. Nov., *M. sinens**e* sp. Nov. and *M. hainanense* sp. Nov., are described with MycoBank number, etymology, typification, morphological features and illustrations, as well as placement on molecular phylogenetic trees. Their affinity with morphologically allied and molecularly closely related species are also analyzed. For facilitating identification, an updated key to the species of *Microdochium* is provided herein.

## 1. Introduction

*Microdochium* Syd. & P. Syd. is a fungal genus in the family *Amphisphaeriaceae* G. Winter of the order *Amphisphaeriales* D. Hawksw. & O.E. Erikss., which was established by Sydow [1] and typified by *M. phragmitis* Syd. & P. Syd. on living leaves of the plant host *Phragmites australis* (Cav.) Trin. ex Steud. This genus is characterized by spherical and erumpent stromata composed of minute and transparent cells, small papilla, conical sporulation cells and solitary transparent spindle-shaped-to-oval conidia. In recent years, many taxonomists have continuously enriched the known diversity in *Microdochium* [2,3,4,5,6,7,8]. Currently, 54 names are listed for this genus in the Index Fungorum [9], but only 37 species are accepted in the Catalogue of Life [10]. They are difficult to cultivate; therefore, just two-fifths have been studied in pure culture [4,5,6,7].

*Microdochium* sensu lato is known to be polyphyletic [2]. While one species, *M. oryzae* (Hashioka & Yokogi) Samuels & I.C. Hallett, was synonymized with *M. albescens* (Thüm.) Hern.-Restr. & Crous [2], and one species, *M. sorghi* (D.C. Bain & Edgerton ex Deighton) U. Braun, was recognized as a synonym of its basionym *Gloeocercospora sorghi* D.C. Bain & Edgerton ex Deighton [9,10], seven species were reclassified to other genera [2,11,12]. In detail, *M. dimerum* (Penz.) Arx, *M. falcatum* B. Sutton & Hodges, *M. fusarioides* D.C. Harris, *M. gracile* Mouch. & Samson, *M. lunatum* (Ellis & Everh.) Arx, *M. tabacinum* (J.F.H. Beyma) Arx, and *M. tripsaci* were transferred to genera *Bisifusarium* L. Lombard, Crous & W. Gams, *Idriella* P.E. Nelson & S. Wilh., *Hyalorbilia* Baral & G. Marson, *Paramicrodochium* Hern-Restr. & Crous, *Bisifusarium*, *Plectosphaerella* Kleb., and *Ephelis* Fr., respectively [2,13,14,15,16,17,18]. Currently, *Microdochium* sensu stricto is a monophyletic clade. A phylogenetic analysis of translation elongation factor 1-alpha gene (TEF1) showed that the isolates of *M. nivale* (Fr.) Samuels & I.C. Hallett were heterogeneous, and hence the variety *M. nivale* var. *majus* (Wollenw.) Samuels & I.C. Hallett was raised to a species rank as *M. majus* (Wollenw.) Glynn & S.G. Edwards, which was still thought to be sister to *M. nivale* [11].

*Microdochium* is an important plant pathogen in grasses and cereals. Liang et al. [19] identified *M. poae* J.M. Liang & Lei Cai as pathogen of *Poa pratensis* L. (Kentucky bluegrass) and *Agrostis stolonifera* L. (creeping bentgrass) which are both cold-season turfgrasses and widely grown on golf courses in northern China. In cold temperate regions, *M. nivale* (=*M. nivale* var. *nivale*) and *M. majus* (=*M. nivale* var. *majus*) [11,12,20] cause “Microdochium patch” on wheat and barley, resulting in significant economic losses. Some species of *Microdochium* are Brassicaceae-associated endophytes in low-Pi conditions (2.48 mg/L) and low-pH conditions (3.4–4.4) [21,22], and *M. bolleyi* is found to be endophytically associated with plant shoots and roots [23] and further to be biocontrol-active against *Gaeumannomyces graminis* var. *tritici* Walker, which causes barley’s take-all disease [24]. A few species are fungicolous, such as *M. fusarioides* D.C. Harris on the oospore of *Phytophthora syringae* (Kleb.) Kleb [17].

In this study, three new pathogenic species in *Microdochium* were found among samples collected in Hainan Island, China. Two of them were isolated from *Miscanthus sinensis* Anderss, and a third one from *Phragmites australis* (Cav.) Trin. ex Steud. Their morphological characteristics and molecular-sequence data are described and discussed below.

## 2. Materials and Methods

### 2.1. Isolation and Morphology

Samples were collected from Hainan Province, China (108°37′–117°50′ E, 3°58′–20°20′ N). The strains of *Microdochium* were isolated from diseased leaves of *Miscanthus sinensis* and *Phragmites australis* using a tissue-isolation method [25]. Tissue fragments (5 × 5 mm) were taken from the margin of leaf lesions and surface-sterilized by immersing consecutively in 75% ethanol solution for 1 min, 5% sodium hypochlorite solution for 30 s, and then rinsing in sterile distilled water for 1 min [26,27]. The sterilized leaf fragments were dried with sterilized paper towels and placed on potato-dextrose agar (PDA) [28]. All the plates were incubated in a biochemical incubator at 25 °C for 3–4 days, after which hyphae were picked out of the periphery of the colonies and transferred onto new PDA plates and oatmeal-agar (OA) [29] plates.

Pure cultures transferred to PDA and OA plates were incubated at 25 °C for 15 days and photographed twice at the 7th and 15th days using a Powershot G7X mark II digital camera. Macro- and micromorphological characteristics were observed using an Olympus SZX10 stereomicroscope and an Olympus BX53 light microscope, respectively. These two microscopes were both fitted with an Olympus DP80 high-definition color digital camera to photo-document fungal structures. All fungal strains were preserved at 4 °C in sterilized 10% glycerin for further studies. Voucher specimens were deposited in the Herbarium Mycologicum Academiae Sinicae, Institute of Microbiology, Chinese Academy of Sciences, Beijing, China (HMAS) and Herbarium of the Department of Plant Pathology, Shandong Agricultural University, Taian, China (HSAUP).

Living cultures were deposited in the Shandong Agricultural University Culture Collection (SAUCC). Taxonomic information on the new taxa was submitted to MycoBank (http://www.mycobank.org/, accessed on 25 April 2022).

### 2.2. DNA Extraction and Amplification

Genomic DNAs were extracted from fungal mycelia grown on PDA, using a modified cetyltrimethylammonium bromide (CTAB) protocol as described in Guo et al. [30]. Four pairs of primers were adopted to amplify four genetic markers [2]. Partial nuclear ribosomal large subunit (LSU), entire internal transcribed spacer (ITS) of rDNA, partial beta-tubulin gene (TUB2), and partial RNA polymerase II second-largest subunit (RPB2) were amplified and sequenced using primer pairs LR0R/LR5 [31], ITS4/ITS5 [32], Btub526F and Btub1332R [12], and RPB2-5F2/fRPB2-7cR [33,34], respectively.

PCRs were performed using an Eppendorf Master Thermocycler (Hamburg, Germany). Amplification reactions were carried out in a volume of 25 μL, containing 12.5 μL 2 × Green Taq Mix (Vazyme, Nanjing, China), 1 μL of each forward and reverse primers (10 μM) (Biosune, Shanghai, China), 1 μL of template genomic DNA (approximately 10 ng/μL), and 9.5 μL of distilled deionized water.

The PCR program consisted of an initial denaturation at 94 °C for 5 min, 35 cycles × [denaturation at 94 °C for 30 s, annealing at a suitable temperature for 30 s, extension at 72 °C for 1 min] and a final elongation at 72 °C for 10 min. Annealing temperatures were 55 °C for ITS, 51 °C for LSU, 56 °C for RPB2 and 53 °C for TUB2. PCR products were visualized through 1% agarose-gel electrophoresis. Paired-end sequencing was conducted by Biosune Company Limited (Shanghai, China). Sequences were proofread for basic authenticity and reliability according to the five simple guidelines established by Nilsson et al. [35]. Consensus sequences were obtained using MEGA 7.0 [36]. All sequences generated in this study were deposited in GenBank (Table 1).

### 2.3. Phylogenetic Analyses

Twenty-eight new sequences were generated in this study, and available reference sequences of *Microdochium* species were retrieved from GenBank [2,3,4,5,6,7]. Four genetic markers (ITS, LSU, TUB2 and RPB2) were separately aligned using MAFFT v.7.110 (Osaka, Japan) [37]. Phylogenetic analyses were conducted individually for each marker at first and then for a combined dataset of the four genetic markers (Appendix A).

Phylogenetic analyses were conducted with Bayesian inference (BI) and maximum-likelihood (ML) algorithms on the CIPRES Science Gateway portal (https://www.phylo.org/, accessed on 15 April 2022;) [38]. The BI ran with MrBayes on XSEDE v. 3.2.7a (Stockholm, Sweden) [39,40,41], and the ML ran with RAxML-HPC2 on XSEDE v. 8.2.12 (Heidelberg, Germany) [42]. The best evolutionary model for each partition was determined using MrModelTest v. 2.3 [43]. Default parameters were used for 1000 bootstrap ML analysis. In BI analysis, starting trees were random, and four MCMC chains ran simultaneously for five million generations. Trees were sampled once every 500 generations. These chains stopped when all convergences met and the standard deviation fell below 0.01. The burn-in fraction was set to 0.25 and Posterior Probabilities (PP) were determined from the remaining trees. All resulting trees were plotted using FigTree v. 1.4.4 (http://tree.bio.ed.ac.uk/software/figtree, accessed on 15 April 2022) and the layout of the trees was carried out with Adobe Illustrator CC 2019.

## 3. Results

### 3.1. Phylogenetic Analyses

Seven *Microdochium* strains isolated from plant hosts were sequenced. Multilocus data (ITS, LSU, TUB2 and RPB2) were composed of 52 strains of *Microdochium* as ingroup and a strain CBS 204.56 of *Idriella lunata* as outgroup. A total of 2957 characters were fed to the phylogenetic analysis, viz. 1–573 (ITS), 574–1423 (LSU), 1424–2117 (TUB2), and 2118–2957 (RPB2). Of these characters, 2223, 97 and 637 were constant, variable parsimony-uninformative and parsimony-informative, respectively. For the BI and ML analyses, the evolutionary model of GTR+I+G was selected for ITS, TUB2 and RPB2, while SYM+I+G was selected for LSU (Figure 1). The topology of the phylogenetic tree generated by the ML method was highly similar to that by BI, and therefore it was chosen to represent the evolutionary history of *Microdochium*.

The 59 strains are assigned to 29 species clades based on the four-marker phylogeny (Figure 1). The seven strains isolated herein represent three novel species. The new species *M. miscanthi* (SAUCC211092, SAUCC211093 and SAUCC211094) has a sister relationship to another new species, *M. sinense* (SAUCC211097 and SAUCC211098), with robust support values (BIPP 1.00 and MLBV 100%). These two species are closely related to *M. rhopalostylidis* (CBS 145125), *M. phragmitis* (CBS 285.71 and CBS 423.78), *M. lycopodinum* (CBS 146.68, CBS 109397 and CBS 109398), *M. indocalami* (SAUCC1016) and *M. fisheri* (CBS 242.90) with high support values (BIPP 1.00 and MLBV 100%). The last new species, *M. hainanense* (SAUCC210781 and SAUCC210782), forms the sister group of the seven species mentioned above with reasonable support (MLBV 92%).

### 3.2. Taxonomy

*Microdochium miscanthi* (Figure 2) S.B. Liu, X.Y. Liu, Z. Meng & X.G. Zhang, sp. nov. 

MycoBank No.: 843867

Etymology—The epithet “*miscanthi*” refers to the genus name of the host plant *Miscanthus sinensis*.

Type—China, Hainan Province: Diaoluoshan National Forest Park, on diseased leaves of *Miscanthus sinensis*, 21 May 2021, S.B. Liu, holotype HMAS352151, isotype HSAUP211092, ex-holotype living culture SAUCC211092.

Description—Colonies on PDA at 25 °C for 14 days attain 87.2–89.1 mm in diameter. When young, round in shape, dark green in the center and white at the edge, with some dark green parts covered with continuously growing mycelia. When old, tight, uneven and pale yellow in the center, fluffy, flat, white at the edge. Mycelia are superficial and immersed, 1.5–2.3 µm wide, transparent, branched and diaphragmatic. Conidiophores are straight or slightly curved, produced on aerial mycelia, septate and often reduced to conidiogenous cells borne directly from hyphae. Conidiogenous cells are mono- or polyblastic, terminal, denticulate, transparent, smooth and cylindrical, 9.7–14.5 × 3.6–4.1 µm. Conidia are solitary, transparent, spindle-to-rod-shaped, 0–2-septate, 7.0–16.1 × 2.5–4.7 µm, 0–5 guttulate when mature and sometimes borne directly from hyphae. Chlamydospores were not observed. Sexual morphs unknown.

Culture characteristics—Colonies on OA at 25 °C for 14 days, reach 88.4–89.3 mm in diameter, and are circular, black-green in the center and irregular in shape, covered with a thin layer of white mycelia, dense at the edge and forming a white ring. Substrate hyphae are transparent and smooth. Vegetative hyphae are transparent, smooth, branched and diaphragmatic.

Notes—Strains SAUCC211092, SAUCC211093 and SAUCC211094 are identified as the same new species *Microdochium miscanthi*. They have similar morphological characteristics, including culture characteristics, sporodochia and conidia. They are also the same in DNA sequences, gathering together with robust support values (MLBV 100% and BIPP 1.00, Figure 1). Phylogenetic analyses on a combined dataset of four genetic markers showed that *M. miscanthi*, *M. lycopodinum*, *M. phragmites*, *M. rhopalostylidis*, *M. fisheri* and *M. sinense* formed a clade. *M. miscanthi* and *M. sinense* form sister clades on the phylogenetic trees, but they are different in culture characteristics, conidia and DNA sequences. In *M. miscanthi*, colonies on PDA are overall white, with central dark-green plaque covered by white mycelia; conidiogenous cells are 9.7–14.5 × 3.6–4.1 µm, without diaphragms; conidia are 7.0–16.1 × 2.5–4.7 µm, spindle-to-rod-shaped. In *M. sinense*, colonies are overall pale yellow; conidiogenous cells are 6.3–22.4 × 4.1–5.7 µm, with single or multiple diaphragms; conidia are 11.5–19.34 × 2.8–5.4 µm, spindle-shaped or cylindrical. As for molecular differences between *M. miscanthi* and *M. sinense*, ITS, BTUB, LSU and RPB2 had 10, 21, 2 and 35 bp of dissimilarity, respectively. Therefore, we assign them in two different species. In addition, conidiogenous cells in *M. miscanthi* are terminal or sympodial, denticulate, transparent, smooth and cylindrical, which are similar to the species in this clade. The conidia of *M. miscanthi* (7.0–16.1 × 2.5–4.7 µm) differs in size from those of *M. lycopodinum* (8.0–15.5 × 2.5–4.0 µm), *M. phragmites* (10.0–14.5 × 2.0–3.0 µm), *M. fisheri* (7.0–12.0 × 3.0–4.0 µm) and *M. rhopalostylidis* (16.0–20.0 × 3.0–4.0 µm) [2,5]. Furthermore, mature conidia are guttulate in *M. miscanthi*.

*Microdochium sinens**e* (Figure 3) S.B. Liu, X.Y. Liu, Z. Meng & X.G. Zhang, sp. nov.

MycoBank—No: 843868

Etymology—The epithet “*sinense*” (Lat.) refers to China, where the species was collected.

Type—China, Hainan Province: *Diaoluoshan* National Forest Park, on diseased leaves of *Miscanthus sinensis*, 21 May 2021, S.B. Liu, holotype HMAS352154, isotype HSAUP211097, ex-holotype living culture SAUCC211097.

Description—Colonies on PDA at 25 °C for 14 days attain 87.2–89.3 mm in diameter; when young, they are irregular in shape, dark green in the center and covered by white hyphae; when old, they are dark green overall, covered completely by white, lush, fluffy and beige hyphae. Mycelia are superficial and immersed, 1.3–2.3 µm wide, transparent, branched and diaphragmatic. Conidiophores are straight or slightly curved, produced from aerial hyphae, septate and often reduced to conidiogenous cells borne directly from hyphae. Conidiogenous cells are monoblastic, terminal, hyaline, smooth and cylindrical, 16.3–22.4 × 4.1–5.7 µm. Conidia are solitary, hyaline, spindle-shaped or cylindrical, 1–3-septate, 11.5–19.34 × 2.8–5.4 µm, 2–9 guttulate when mature and sometimes borne directly from hyphae. Chlamydospores were not observed. Sexual morphs unknown.

Culture characteristics—Colonies on OA at 25 °C for 14 days, reach 86.4–88.9 mm in diameter; when young, they are circular gray in the center and wax yellow at the edge; when old, they have ravines, dense, yellow-brown overall and fluffy at the edge. Vegetative hyphae are transparent, branched and diaphragmatic.

Notes—Strains SAUCC211097 and SAUCC211098 are identified to the same species *Microdochium sinense* sp. nov. For details, refer to the notes for *M. miscanthi*.

*Microdochium hainanense* (Figure 4) S.B. Liu, X.Y. Liu, Z. Meng & X.G. Zhang, sp. nov.

MycoBank—No:843869

Etymology—The epithet “*hainanense*” is named after Hainan Province, where the fungus was collected.

Type—China, Hainan Province: Diaoluoshan National Forest Park, on diseased leaves of *Phragmites australis*, 21 May 2021, S.B. Liu, holotype HMAS352156, isotype HSAUP210781, ex-holotype living culture SAUCC210781.

Description—Colonies on PDA for 14 days attain 75.4–77.2 mm in diameter; when young, they form a conspicuously concentric circle, brown and dense in the center, white and sparse at the edge; when old, they produce sporodochia in aerial mycelia or on agar surface, slimy, hyaline or orange, colorless-to-brownish in reverse due to secreted soluble pigments. Mycelia are superficial and immersed, width 1.5–3.0 µm, transparent, smooth, branched and diaphragmatic. Conidiophores are reduced to conidiogenous cells. Conidiogenous cells are monoblastic, terminal, hyaline, smooth, ampulliform and lageniform, with percurrent proliferations, 4.8–8.2 × 2.0–2.5 µm. Conidia are solitary, hyaline, aseptate and spindle-to-rod-shaped, 7.0–16.1 × 2.5–4.7 µm, 0–8 guttulate when mature. Chlamydospores were not observed. Sexual morphs unknown.

Culture characteristics—Colonies on OA at 25 °C for 14 days reach 69.7–71.9 mm in diameter; they are circular, with hyphae mostly immersed in agar and occasionally scattered on the agar surface; light black and sparse in the center, white and dense at the edge. Substrate hyphae are transparent and smooth. Vegetative hyphae are transparent, septate and branched.

Notes—Strains SAUCC210781 and SAUCC210782 are identified to the same new species, *M. hainanense*. They share morphological characteristics, including culture characteristics, sporodochia and conidia. They are also identical in DNA sequences, gathering together with robust support values (MLBV 100% and BIPP 1.00, Figure 1). Phylogenetic analysis of the four genetic markers of *M. hainanense* showed that *M. hainanense* formed an independent branch, sister to the group of *M. indocalami*, *M. sinense*, *M. miscanthi*, *M. rhopalostylidis*, *M. phragmites, M. fisheri* and *M. lycopodinum* with satisfactory support (MLBV 92%, Figure 1). *Microdochium. hainanense* produces sporodochia, similar to *M. phragmitis* (CBS 423.78) and *M. rhopalostylidis*, but *M. hainanense* produces clear-to-orange soluble pigments, while the conidia of other species are directly produced from hyphae. Conidia are single, ellipsoid or spindle-shaped, similar to all the related species mentioned above. Conidia of *M. hainanense* (5.5–8.1 × 2.2–3.0 µm) differ in size from those of *M. lycopodinum* (8.0–15.5 × 2.5–4.0 µm), *M. phragmites* (10.0–14.5 × 2.0–3.0 µm), *M. rhopalostylidis* (16.0–20.0 × 3.0 –4.0 µm), *M. indocalami* (13.0–15.5 × 3.5–5.5 µm), *M. fisheri* (7.0–12.0 × 3.0–4.0 µm), *M. miscanthi* (7.0–16.1 × 2.5–4.7 µm) and *M. sinense* (11.5–19.34 × 2.8–5.4 µm) [2,5].

### 3.3. Key to the Species of Microdochium

Together with the three new species proposed in this study, we currently accepted a worldwide total of 47 species in the genus *Microdochium*. In order to facilitate identification in the future, a key to the species of *Microdochium* is provided herein, updating the key compiled 46 years ago [15]. Characteristics adopted in the key include perithecia, septa, asci, ascospores, conidiogenous cells, conidia and chlamydospores.

1. Sexual morph known------------------------------------------------------------------------------------21. Sexual morph unknown-------------------------------------------------------------------------------132. Perithecia maximum diameter > 200 μm------------------------------------------------------------32. Perithecia maximum diameter < 200 μm------------------------------------------------------------93. Maximum number of septa of ascospores > 3----------------------------------------------------- 43. Maximum number of septa of ascospores ≤ 3------------------------------------------------------54. Asci size 90.0–120.0 × 21.0–25.0 μm-------------------------------------------------*M. consociatum*4. Asci size 80.0–100.0 × 17.0–22.0 μm--------------------------------------------------------*M. musae*5. Asci size = 50.0–70.0 × 7.0–9.0 μm---------------------------------------------------------------------65. Asci size ≠ 50.0–70.0 × 7.0–9.0 μm---------------------------------------------------------------------76. Ascospores size 9.5–17.0 × 3.0–4.5 μm-----------------------------------------------------*M. majus*6. Ascospores size 10.0–17.0 × 3.5–4.5 μm----------------------------------------------------*M. nivale*7. Ascospores 1–3 septa-------------------------------------------------------------------------------------87. Ascospores 1–2 septa--------------------------------------------------------------------*M. stevensonii*8. Ascospores size 20.0–32.0 × 3.0–3.5 μm------------------------------------------*M. fusariisporum*8. Ascospores size 15.0–25.0 × 4.0–5.0 μm----------------------------------------------*M. passiflorae*9. Perithecia maximum diameter < 150 μm----------------------------------------------------------109. Perithecia maximum diameter > 150 μm----------------------------------------------------------1110. Ascospores size 20.0–22.0 × 3.5 μm----------------------------------------------------*M. opuntiae*10. Ascospores size 12.0–22.0 × 3.0–5.0 μm---------------------------------------------*M. seminicola*11. Chlamydospores known--------------------------------------------------------------*M. ratticaudae*11. Chlamydospores unknown--------------------------------------------------------------------------1212. Conidia falcate, 11.0–16.0 × 3.5–4.5 μm, 0–3 septa---------------------------------*M. albescens*12. Conidia lunate, 8.0–15.0 × 2.5–3.5 μm, 0–1 septa------------------------------*M. lycopodinum*13. Chlamydospores known-----------------------------------------------------------------------------1413. Chlamydospores unknown--------------------------------------------------------------------------1614. Conidia oblong---------------------------------------------------------------------*M. trichocladiopsis*14. Conidia lunate------------------------------------------------------------------------------------------1515. Chlamydospores chain or clusters--------------------------------------------------------*M. bolleyi*15. Chlamydospores rounded or obovoid-----------------------------------------------------*M. poae*16. Conidia aseptate----------------------------------------------------------------------------------------1716. Conidia septate-----------------------------------------------------------------------------------------2317. Conidiogenous cells two types-----------------------------------------------------*M. yunnanense*17. Conidiogenous cells one type-----------------------------------------------------------------------1818. Conidiogenous cells with denticulate-------------------------------------------------------------1918. Conidiogenous cells not denticulate--------------------------------------------------------------2119. Conidiogenous cells ampulliform------------------------------------------------------------------2019. Conidiogenous cells cylindrical----------------------------------------------------*M. sclerotiorum*20. Conidia pointed at both ends, no appendages--------------------------------------*M. griseum*20. Conidia with straight appendages at both ends---------------------------*M. queenslandicum*21. Conidiogenous cells monoblastic--------------------------------------***M. hainanense* sp. nov.**21. Conidiogenous cells sympodial--------------------------------------------------------------------2222. Conidia filiform, 7.0–16.0 × 1.0 μm----------------------------------------------------*M. palmicola*22. Conidia lunate, 7.5.0–11.0 × 1.8–2.0 μm--------------------------------------*M. queenslandicum*23. Conidiogenous cells two types------------------------------------------------------*M. colombiense*23. Conidiogenous cells one type-----------------------------------------------------------------------2424. Conidia relatively narrow, acicular, filiform, falcate or lunate-----------------------------2524. Conidia relatively rounded, ellipsoid, fusiform, cylindrical or obovoid-----------------3225. Conidia with long appendages at both ends------------------------------------------*M. linariae*25. Conidia without appendages at both ends------------------------------------------------------2626. Conidia with conspicuous rhachides-----------------------------------------------*M. tainanense*26. Conidia without conspicuous rhachides---------------------------------------------------------2727. Conidiogenous cells ampulliform------------------------------------------------------------------2827. Conidiogenous cells cylindrical--------------------------------------------------------------------3128. Maximum number of septa of conidia = 10---------------------------------------------*M. sorghi*28. Maximum number of septa of conidia < 10------------------------------------------------------2929. Conidia lunate-----------------------------------------------------------------*M. neoqueenslandicum*29. Conidia falcate------------------------------------------------------------------------------------------3030. Conidia size 25.0–30.0 × 1.5–2.0 μm, 0–1 septa----------------------------------*M. caespitosum*30. Conidia size 7.0– 20.5 ×2.5–4.5 μm, 0–3 septa-----------------------------------------*M. paspali*31. Conidia size 25.0–75.0 × 1.0–2.0 μm, 0–3 septa--------------------------------*M. dawsoniorum*31. Conidia size 5.5–10.0 × 2.0–2.5 μm, 0–1 septa-------------------------------*M. novae-zelandiae*32. Conidia with guttulate--------------------------------------------------------------------------------3332. Conidia no guttulate----------------------------------------------------------------------------------3533. Conidiogenous cells solitary-------------------------------------------------*M. chrysanthemoides*33. Conidiogenous cells sympodial--------------------------------------------------------------------3434. Conidia size 10.0–14.5 × 2.0–3.0 μm, 0–1 septa-----------------------------------*M. phragmitis*34. Conidia size 13.0–23.0 × 2.5–4.0 μm, 1–3 septa------------------------------*M. rhopalostylidis*35. Conidia cylindrical-------------------------------------------------------------------------------------3635. Conidia fusiform---------------------------------------------------------------------------------------4236. Conidiogenous cells denticulate--------------------------------------------------------------------3736. Conidiogenous cells not denticulate--------------------------------------------------------------4037. Conidiogenous cells blastic-sympodial-------------------------------------------*M. cylindricum*37. Conidiogenous cells mono- or polyblastic------------------------------------------------------3838. Conidia spindle-to-rod-shaped-------------------------------------------***M. miscanthi* sp. nov.**38. Conidia clavate to obovoid--------------------------------------------------------------------------3939. Conidia size 7.0–31.0 × 2.0–3.0 μm, 0–3 septa----------------------------------*M. citrinidiscum*39. Conidia size 13.0–15.5 × 3.5–5.5 μm, 1–3 septa-----------------------------------*M. indocalami*40. Conidiogenous cells ampulliform------------------------------------------------------*M. maydis*40. Conidiogenous cells cylindrical--------------------------------------------------------------------4141. Conidiogenous cells monoblastic, 16.3–22.4 × 4.1–5.7 μm-------------***M. sinense* sp. nov.**41. Conidiogenous cells sympodial, 6.5–15.0 × 2.5–3.5 μm-----------------------------*M. stoveri*42. Conidiogenous cells ampulliform------------------------------------------------------------------4342. Conidiogenous cells cylindrical--------------------------------------------------------------------4443. Conidiogenous cells solitary------------------------------------------------------------*M. punctum*43. Conidiogenous cells sympodial-------------------------------------------------------*M. triticicola*44. Conidiogenous cells mono- or polyblastic---------------------------------------*M. maculosum*44. Conidiogenous cells sympodial--------------------------------------------------------------------4545. Conidiogenous cells not denticulate------------------------------------------*M. panattonianum*45. Conidiogenous cells denticulate--------------------------------------------------------------------4646. Conidia size 7.0–12.0 × 3.0–4.0 μm, 0–1 septa------------------------------------------*M. fisheri*46. Conidia size 8.0–15.0 × 3.0–4.5 μm, 1–2 septa----------------------------------*M. intermedium*

## 4. Discussion

*Microdochium* was established in 1924, and *Monographella* Petr. also established in 1924 was previously described as a sexual morph of *Microdochium* [16,44,45,46]. With the application of “one fungus one name” declaration [47], *Microdochium* was retained as the correct genus name because it accommodates more species and is used more frequently [2]. Due to their phylogenetic affinity, *Microdochium*, *Idriella* and *Selenodriella* were introduced into a new family, namely *Microdochiaceae* [2]. This new family is characterized by (1) *Monographella*-like sexual morphs; and (2) asexual morphs of polyblastic, sympodial or annellidic conidiogenous cells with hyaline conidia, but no appendages. As an important basis for classification, conidia of *Microdochium* vary in shape, i.e., cylindrical, fusiform, elliptical, stick-shaped, vertical or curved, with truncate bases and apices mainly rounded.

Since the inception of *Microdochium* in 1924, its delimitation has undergone changes, and currently, 47 species are accepted in the genus. Although the number is small, there are still some problems in the classification. For example, Catalogue of Life accepts the basionym *Gloeocercospora sorghi* rather than the combination *Microdochium sorghi* but without any explanation [10]. It is possibly because *M. sorghi* remains sterile and only produces black sclerotia in culture [2]. However, in this study, the phylogenetic analysis based on the based on four genetic markers showed that *M. sorghi* formed a separated branch closely related to the clade of *M. citrinidiscum* and *M. paspali* with strong support (MLBV: 100% and BIPP: 1.00, Figure 1). Upon this molecular evidence, we accept *M. sorghi* as the correct name for this species.

*Microdochium* is mainly distributed in warm and humid areas, and most prefer to parasitize on gramineous plants. Our finding of the new species *M. miscanthi*/*M. sinens**e* on *Miscanthus sinensis* (*Poaceae*) and *M. hainanense* on *Phragmites australis* (*Poaceae*), confirms this phenomenon well. Hainan Province is located in the tropical region of southern China. Its annual average temperature is 22–27 °C, and its annual precipitation is 1000–2600 mm, with a typical tropical rainforest climate. This kind of environment is conducive to the growth of unusual microbial species, resulting in a high species diversity.

In order to accurately identify the species of *Microdochium*, molecular analysis is needed. In this study, the four genetic markers ITS, LSU, RPB2 and TUB2 were selected according to previous molecular studies of *Microdochium*. LSU provides enough information for the generic placement of *Microdochium*. Although any of the genetic markers ITS, TUB2 or RPB2 can be used for phylogenetic analysis at the species level in *Microdochium* (results not shown), TUB2 has more phylogenetic information, with longer distances between species and higher support values. This is consistent with previous studies on other xylariaceous genera [2,48,49].

## Figures and Tables

**Figure 1 jof-08-00577-f001:**
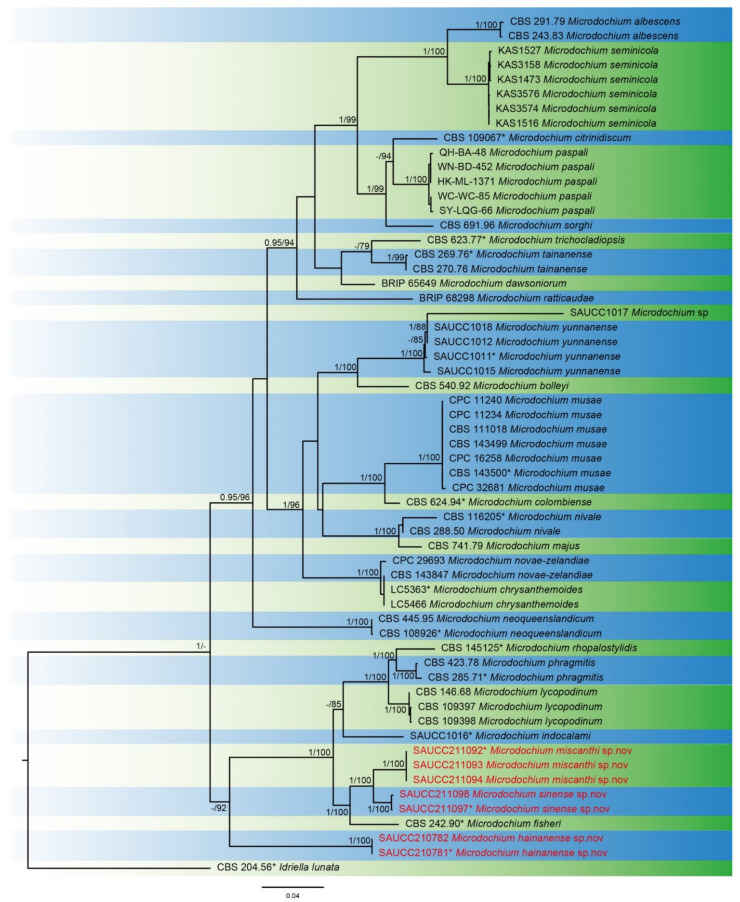
A maximum-likelihood phylogram of *Microdochium* based on combined ITS, LSU, TUB2 and RPB2 sequences with CBS 204.56 of *Idriella lunata* as outgroup. The maximum-likelihood Bootstrap Value (MLBV ≥ 75%) and Bayesian Inference Posterior Probability (BIPP ≥ 0.95) are shown at the first and second position, respectively. Strains marked with “*” are ex-types, ex-epitypes or holotypes. Strains from the current study are in red. The scale bar at the bottom middle indicates 0.08 substitutions per site.

**Figure 2 jof-08-00577-f002:**
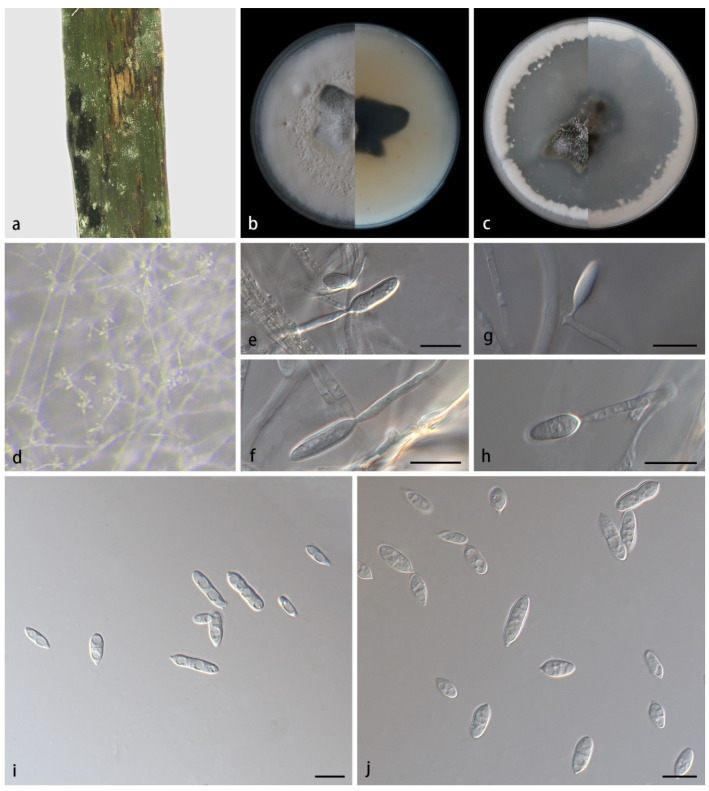
*Microdochium miscanthi* (holotype HMAS352151, ex-holotype SAUCC211092). (**a**) Leaves of host plant; (**b**) inverse and reverse sides of colony after 15 days on PDA; (**c**) inverse and reverse sides of colony after 15 days on OA; (**d**) a colony overview; (**e**–**h**) conidiophores and conidiogenous cells; (**i**,**j**) conidia. Scale bars: (**e**–**j**) 10 μm.

**Figure 3 jof-08-00577-f003:**
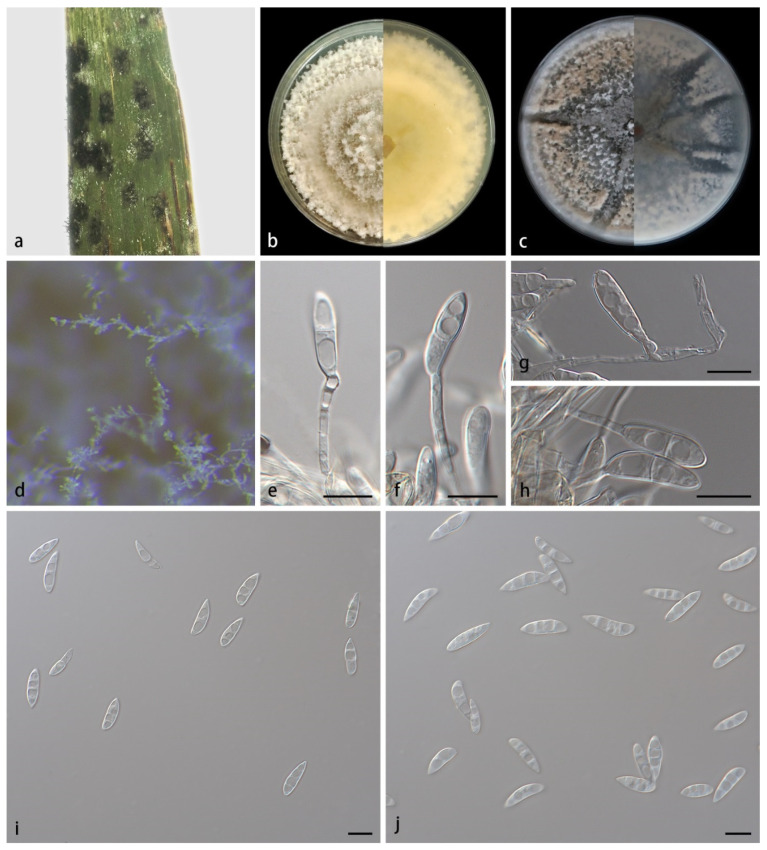
*Microdochium sinense* (holotype HMAS352154, ex-holotype SAUCC211097). (**a**) Leaves of host plant; (**b**) inverse and reverse sides of colony after 15 days on PDA; (**c**) inverse and reverse sides of colony after 15 days on OA; (**d**) colony overview; (**e**–**h**) conidiophores and conidiogenous cells; (**i**,**j**) conidia. Scale bars: (**e**–**j**) 10 μm.

**Figure 4 jof-08-00577-f004:**
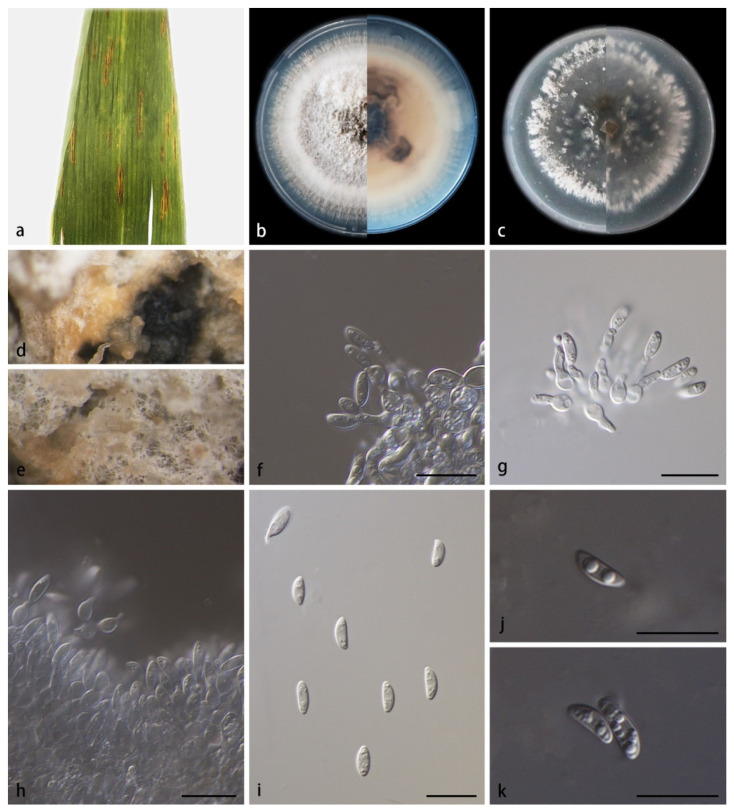
*Microdochium hainanense* (holotype HMAS352156, ex-holotype SAUCC210781). (**a**) leaves of host plant; (**b**) inverse and reverse sides of colony after 15 days on PDA; (**c**) inverse and reverse sides of colony after 15 days on OA; (**d**) sporodochia after removing the surface mycelia; (**e**) the mixture of conidia and secretions on mycelium; (**f**–**h**) conidiophores and conidiogenous cells; (**i**–**k**) conidia. Scale bars: (**f**–**k**) 10 μm.

**Table 1 jof-08-00577-t001:** Information of specimens used in this study.

Species	Voucher	Host/Substrate	Country	GenBank Accession Numbers			
				LSU	ITS	BTUB	RPB2
*Idriella. lunata*	CBS 204.56 *	*Fragaria chiloensis*	USA	KP858981	KP859044	–	–
*Microdochium. albescens*	CBS 291.79	*Oryza sativa*	Ivory Coast	KP858932	KP858996	KP859059	KP859105
	CBS 243.83	*Oryza sativa*	Unknown	KP858930	KP858994	KP859057	KP859103
*M. bolleyi*	CBS 540.92	*Hordeum vulgare*	Syria	KP858946	KP859010	KP859073	KP859119
*M. chrysanthemoides*	CGMCC3.17929 *	Unnamed Karst Cave	China	KU746736	KU746690	–	–
	CGMCC3.17930 *	Unnamed Karst Cave	China	KU746735	KU746689	–	–
*M. citrinidiscum*	CBS 109067 *	*Eichhornia crassipes*	Peru	KP858939	KP859003	KP859066	KP859112
*M. colombiense*	CBS 624.94 *	*Musa sapientum*	Colombia	KP858935	KP858999	KP859062	KP859108
*M. dawsoniorum*	BRIP 65649	*Sporobolus*	Australia	–	MK966337	–	–
*M. fisheri*	CBS 242.90 *	*Oryza sativa*	UK	KP858951	KP859015	KP859079	KP859124
** *M. hainanense* **	SAUCC210781 *	*Phragmites australis*	China	OM959323	OM956295	OM981146	OM981153
	SAUCC210782	*Phragmites australis*	China	OM959324	OM956296	OM981147	OM981154
*M. indocalami*	SAUCC1016 *	*Indocalamus longiauritus*	China	MT199878	MT199884	MT435653	MT510550
*M. lycopodinum*	CBS 146.68	Air samples	The Netherlands	KP858929	KP858993	KP859056	KP859102
	CBS 109397	*Phragmites australis*	Germany	KP858940	KP859004	KP859067	KP859113
	CBS 109398	*Phragmites australis*	Germany	KP858941	KP859005	KP859068	KP859114
*M. majus*	CBS 741.79	*Triticum aestivum*	Germany	KP858937	KP859001	KP859064	KP859110
** *M. miscanthi* **	SAUCC211092 *	*Miscanthus sinensis*	China	OM957532	OM956214	OM981141	OM981148
	SAUCC211093	*Miscanthus sinensis*	China	OM957533	OM956215	OM981142	OM981149
	SAUCC211094	*Miscanthus sinensis*	China	OM957534	OM956216	OM981143	OM981150
*M. musae*	CBS 111018 = CPC 5380	*Musa* cv. *cavendish*	Costa Rica	–	AY293061	–	–
	CBS 143499 = CPC 32809	*Musa* sp.	Malaysia	MH107941	MH107894	–	–
	CBS 143500 * = CPC 32689	*Musa* sp.	Malaysia	MH107942	MH107895	–	MH108003
	CPC 11234	*Musa* sp.	Mauritius	MH107943	MH107896	–	–
	CPC 11240	*Musa* sp.	Mauritius	MH107944	MH107897	–	–
	CPC 16258	*Musa* sp.	Mexico	MH107945	MH107898	–	–
	CPC 32681	*Musa* sp.	Malaysia	MH107946	MH107899	–	–
*M. neoqueenslandicum*	CBS 445.95	*Juncus effusus*	The Netherlands	KP858933	KP858997	KP859060	KP859106
	CBS 108926 *	*Agrostis* sp.	New Zealand	KP858938	KP859002	KP859065	KP859111
*M. nivale*	CBS 116205 *	*Triticum aestivum*	UK	KP858944	KP859008	KP859071	KP859117
	CBS 288.50	Unknown	Unknown	MH868135	MH856626	–	–
*M. novae-zelandiae*	CBS 143847	Turf leaves (*Poaceae*)	New Zealand	–	LT990655	LT990608	LT990641
	CPC 29693	Turf leaves (*Poaceae*)	New Zealand	–	LT990656	LT990609	LT990642
*M. paspali*	HK-ML-1371	*Paspalum vaginatum*	China	–	KJ569509	KJ569514	–
	QH-BA-48	*Paspalum vaginatum*	China	–	KJ569510	KJ569515	–
	SY-LQG66	*Paspalum vaginatum*	China	–	KJ569511	KJ569516	–
	WC-WC-85	*Paspalum vaginatum*	China	–	KJ569512	KJ569517	–
	WN-BD-452	*Paspalum vaginatum*	China	–	KJ569513	KJ569518	–
*M. phragmitis*	CBS 285.71 *	*Phragmites australis*	Poland	KP858949	KP859013	KP859077	KP859122
	CBS 423.78	*Phragmites communis*	Germany	KP858948	KP859012	KP859076	KP859121
*M. ratticaudae*	BRIP 68298	introduced giant rat’s tail grasses	Australia	MW481666	MW481661	–	MW626890
*M. rhopalostylidis*	CPC 34449 = CBS 145125 *	*Rhopalostylis sapida*	New Zealand	MK442532	MK442592	–	MK442667
*M. seminicola*	KAS3576 = CBS 139951 *	Maize kernels	Switzerland	KP858974	KP859038	KP859101	KP859147
	KAS1516 = CPC 26001	Grain	Canada	KP858961	KP859025	KP859088	KP859134
	KAS3574 = DAOM 250155	*Maize kernels*	Switzerland	KP858973	KP859037	KP859100	KP859146
	KAS3158 = DAOM 250161	*Triticum aestivum*	Canada	KP858970	KP859034	KP859097	KP859143
	KAS1527 = DAOM 250165	Grain	Canada	KP858966	KP859030	KP859093	KP859139
	KAS1473 = DAOM 250176	*Triticum aestivum*	Canada	KP858955	KP859019	KP859082	KP859128
** *M. sinense* **	SAUCC211097 *	*Miscanthus sinensis*	China	OM959225	OM956289	OM981144	OM981151
	SAUCC211098	*Miscanthus sinensis*	China	OM959226	OM956290	OM981145	OM981152
*M. sorghi*	CBS 691.96	*Sorghum halepense*	Cuba	KP858936	KP859000	KP859063	KP859109
*M. sp. indet.*	SAUCC1017	*Indocalamus longiauritus*	China	MT199879	MT199885	MT435654	–
*M. tainanense*	CBS 269.76 *	*Saccharum officinarum*	Taiwan	KP858945	KP859009	KP859072	KP859118
	CBS 270.76	*Saccharum officinarum*	Taiwan	KP858931	KP858995	KP859058	KP859104
*M. trichocladiopsis*	CBS 623.77 *	*Triticum aestivum*	Unknown	KP858934	KP858998	KP859061	KP859107
*M. yunnanense*	SAUCC1011 *	*Indocalamus longiauritus*	China	MT199875	MT199881	MT435650	MT510547
	SAUCC1012	*Indocalamus longiauritus*	China	MT199876	MT199882	–	MT510548
	SAUCC1015	*Indocalamus longiauritus*	Chima	MT199877	MT199883	MT435652	MT510549
	SAUCC1018	*Indocalamus longiauritus*	Chima	MT199880	MT199886	MT435655	–

Notes: New species established in this study are in bold. Ex-types, ex-epitypes or holotype strains are marked with “*”.

## Data Availability

The sequences from the present study were submitted to the NCBI database (https://www.ncbi.nlm.nih.gov/, accessed on 25 April 2022) and the accession numbers were listed in Table 1.

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
