# Peer review of "Three New Species of Microdochium (Sordariomycetes, Amphisphaeriales) on Miscanthus sinensis and Phragmites australis from Hainan, China"

_jof, 2022, doi:10.3390/jof8060577_

Round 1

Reviewer 1 Report

This paper describes Three new species of Microdochium (Sordariomycetes, phisphaeriales) on Miscanthus sinensis and Phragmites australis from Hainan, China. The information is well presented, although conclusion could be improved. The morphology of nov. sp, which can be distinguished from close species, can be improved. The manuscript can be accepted after minor changes. Additional comments/corrections are on the ms.

Author Response

This paper describes Three new species of Microdochium (Sordariomycetes, phisphaeriales) on Miscanthus sinensis and Phragmites australis from Hainan, China. The information is well presented, although conclusion could be improved. The morphology of nov. sp, which can be distinguished from close species, can be improved. The manuscript can be accepted after minor changes. Additional comments/corrections are on the ms.”

Response: Thanks for your suggestion. In section “Taxonomy”, we annotated on the distinction between new species and their allied species, both morphologically and phylogenetically.

Reviewer 2 Report

This is a nice piece that introduces several new species in Microdochium. I was particularly happy to see that a key to all species in the genus is provided. Good thinking!

The study suffers from major complications with the multiple sequence alignment (see below). Some sequences are reverse complementary, or represent contamination, or represent other genetic markers than the intended one. That means that the phylogenetic tree inferred by the authors is artefactual and cannot be published. See below.

Line 12. ”were reported often” > ”have often been reported as”

  1. ”annotated” > ”analysed”

20-21. Nice to see a key bundled with this paper!

25-26. The authors violate the recommendations of ICTF by not writing fungal family + order names in italics (https://imafungus.biomedcentral.com/articles/10.1186/s43008-020-00048-6 ).

  1. “the species” > “the known”

  1. “was shown” > “is known”

  1. The “usually” can be deleted.

  1. What is “low Pi conditions” ? “low pH conditions” ? Please clarify.

  1. But Phytophthora is an oomycete rather than a fungus. Does the term “fungicolous” still apply?

  1. “from the” > “among”

  1. Please provide GPS coordinates of the type localities of the new species (see https://mycokeys.pensoft.net/articles.php?id=56691&journal_name=mycokeys).

  1. “culture” > “cultures” ; “was” > “were”

  1. “of the” > “on the”

  1. “loci” is a genetics term that is not the same thing as “gene” and that cannot be used like this. “gene” would have worked, but one of the genetic markers targeted by the authors (the ITS region) does not qualify as a “gene”. Thus, “loci” > “genetic markers” here and elsewhere.

  1. “A large” > “Partial nuclear ribosomal large subunit …”

  1. “for genes” should be deleted. Not all of them are genes to begin with, right.

  1. It is standard procedure to evaluate newly generated sequences for basic qualities using, e.g., the guidelines of https://mycokeys.pensoft.net/articles.php?id=1186 . The authors make no mention of having done this, which is worrying.

            And it is particularly troublesome since there are far-reaching problems with the multiple sequence alignment. More on that later.

  1. The authors mark “ex-type” and “ex-epitype” strains in the table. So are there no holotypes at all here, just cultures?

  1. There are far-reaching problems with the multiple sequence alignment.

            To begin with, removed all gap-only columns.

            The alignment looks OK up to around position 889.

            Starting at position 895, sequence number 9 is terribly misaligned. This continues all the way to position 1860. Either the authors sequenced/pasted the wrong genetic marker, or the authors have in fact sequenced some contamination, or it is given in the reverse complementary orientation in the alignment. Please fix this, because right now this part of the alignment is pure noise.

            The same thing (same positions) goes for sequences 35-40.

            The same thing goes for sequence 34 positions 1955-2843.

            The same thing goes for sequences 16-19 positions 1856-2844.

            The same thing goes for sequences 6-7 positions 1871-2856.  

How is it possible that the authors didn’t notice this? Did they not go through the alignment prior to analysis? Please see (https://mycokeys.pensoft.net/articles.php?id=1186 and https://www.creamjournal.org/pdf/Cream_3_1_1.pdf on how to examine the underlying cause of these issues.)

            Bottom line: the alignment must be redone, and so must the phylogenetic analyses. Right now they mainly reflect noise, I would say.

            It’s a good thing that the authors provided the multiple sequence alignment with their manuscript. This is, indeed, what the MIAPA standard dictates (https://pubmed.ncbi.nlm.nih.gov/16901231/).

136-138. What is “This section may be divided by subheadings. It should provide a concise and precise description of the experimental results, their interpretation, as well as the experimental conclusions that can be drawn.” supposed to mean?

140-149. Good work here!

  1. “while” > “while SYM+I+G was selected for”

158 and 274. The word “basal” can not be used like this. See https://resjournals.onlinelibrary.wiley.com/doi/10.1111/j.0307-6970.2004.00262.x - please use the “sister species” construction instead.

  1. “with a sound” > “with reasonable”

  1. The “shortening of branches” has to do with the compromised multiple sequence alignment.

  1. “complied” > “compiled”

  1. The key is a strong asset of this study. Nice work!

  1. “Since” > “Due to”

  1. “birth” > “inception”

  1. “species have” > “delimitation has” or “circumscription has”

  1. I propose: “and currently, 47 species are accepted in the genus.”

  1. “in classification” > “in the classification”

  1. “explanations” > “explanation”. I think the authors should send an email to CoL and ask them to fix this issue. I’m sure the reason is that they are unaware of this complication.

  1. “good” > “strong”

  1. “Fungi in Agricultural Soils” > “Fungi in agricultural soils”. Similarly on line 528.

Author Response

The study suffers from major complications with the multiple sequence alignment (see below). Some sequences are reverse complementary, or represent contamination, or represent other genetic markers than the intended one. That means that the phylogenetic tree inferred by the authors is artefactual and cannot be published. See below.

Response: Thanks for your suggestion. We have dealt with all these issues you mentioned. In detail, we rearranged the sequence, reversely complemented some loci, deleted the contaminated sequence fragments (Annex 1), and in the end reconstructed the phylogenetic tree (Figure 1). Consequently, we revised relevant text in the returned manuscript.

Line 12. “were reported often” > “have often been reported as”

Response: Thank you.

Line 20. “annotated” > “analysed”

Response: Thank you.

Line 20-21. Nice to see a key bundled with this paper!

Response: Thank you for your affirmation.

Line 25-26. The authors violate the recommendations of ICTF by not writing fungal family + order names in italics (https://imafungus.biomedcentral.com/articles/10.1186/s43008-020-00048-6).

Response: Thanks for your suggestion. We have italicized all formal taxonomic taxa in the revised ms.

Line 31. “the species” > “the known”

Response: Thank you.

Line 35. “was shown” > “is known”

Response: Thank you.

Line 51. The “usually” can be deleted.

Response: Thank you. We accepted your suggestion.

Line 58. What is “low Pi conditions”? “low pH conditions”?

Response: Thanks for your suggestion. After referring to the relevant literature, we make clear that the low phosphorus condition means a content of phosphorus is 2.48 mg/L and the low pH condition is 3.4‒4.4. We have provided the revised ms with this information and related references.

Line 61. But Phytophthora is an oomycete rather than a fungus. Does the term “fungicolous” still apply?

Response: Thanks for your suggestion. Here fungicolous refers to a broad meaning, including organisms traditionally studied by mycologists, other than the kingdom Fungi.

Line 63. “from the” > “among”

Response: Thank you.

Line 70. Please provide GPS coordinates of the type localities of the new species (see https://mycokeys.pensoft.net/articles.php?id=56691&journal_name=mycokeys).

Response: Thanks for your suggestion. The GPS coordinates for the new species have been added to the revised ms.

Line 80. “culture” > “cultures” ; “was” > “were”

Response: Thank you.

Line 89. “of the” > “on the”

Response: Thank you.

Line 95. “loci” is a genetics term that is not the same thing as “gene” and that cannot be used like this. “gene” would have worked, but one of the genetic markers targeted by the authors (the ITS region) does not qualify as a “gene”. Thus, “loci” > “genetic markers” here and elsewhere.

Response: Thanks for your suggestion. We accepted and consequently make changes throughout the ms.

Line 95. “A large” > “Partial nuclear ribosomal large subunit …”

Response: Thank you.

Line 107. “for genes” should be deleted. Not all of them are genes to begin with, right

Response: Thanks. We follow your suggestion, and deleted “for genes”.

Line 112. It is standard procedure to evaluate newly generated sequences for basic qualities using, e.g., the guidelines of https://mycokeys.pensoft.net/articles.php?id=1186. The authors make no mention of having done this, which is worrying.

And it is particularly troublesome since there are far-reaching problems with the multiple sequence alignment. More on that later.

Response: Thanks for your suggestion. We proofread the sequences according to 5 simple guidelines established by Nilsson et al. for the basic authenticity and reliability of newly generated fungal ITS sequences, and cited Henrick’s work in the revised ms.

Line 115. The authors mark “ex-type” and “ex-epitype” strains in the table. So are there no holotypes at all here, just cultures?

Response: Thanks for your suggestion. We included the information about the holotype in Table1 and Figure 1 by notes. Really, all molecular sequences are obtained from living ex-types, ex-epitypes and ex-holotypes, rather than types, epitypes or holotypes.

Line 132.There are far-reaching problems with the multiple sequence alignment.

To begin with, removed all gap-only columns.

The alignment looks OK up to around position 889.

Starting at position 895, sequence number 9 is terribly misaligned. This continues all the way to position 1860. Either the authors sequenced/pasted the wrong genetic marker, or the authors have in fact sequenced some contamination, or it is given in the reverse complementary orientation in the alignment. Please fix this, because right now this part of the alignment is pure noise.

The same thing (same positions) goes for sequences 35-40.

The same thing goes for sequence 34 positions 1955-2843.

The same thing goes for sequences 16-19 positions 1856-2844.

The same thing goes for sequences 6-7 positions 1871-2856.

How is it possible that the authors didn’t notice this? Did they not go through the alignment prior to analysis? Please see (https://mycokeys.pensoft.net/articles.php?id=1186 and https://www.creamjournal.org/pdf/Cream_3_1_1.pdf on how to examine the underlying cause of these issues.)

Bottom line: the alignment must be redone, and so must the phylogenetic analyses. Right now they mainly reflect noise, I would say.

It’s a good thing that the authors provided the multiple sequence alignment with their manuscript. This is, indeed, what the MIAPA standard dictates (https://pubmed.ncbi.nlm.nih.gov/16901231/).

Response: Thanks for your suggestion. We double-checked the quality of the sequences following the method of  Nilsson et al. and found that the LUS sequences KP858951, KP858974, KP858961, KP858973, KP858970, KP858966 and KP858955 were reversed, and the TUB sequences KU746781, KU746782, MH108040, MH1080041, MH108041, MH108043, MH108044 and MK442735 contain a large number of introns and a small number of exons. These two reasons cause sequence confusion. Therefore, we reverse-complement questioned LSU sequence and delete the TUB sequence with too many introns to solve this problem. In the end, we reconstructed the phylogenetic tree, and revised the "Results" section of the ms.

Line 136-138. What is “This section may be divided by subheadings. It should provide a concise and precise description of the experimental results, their interpretation, as well as the experimental conclusions that can be drawn.” supposed to mean?

Response: Thanks for your suggestion. We apologize about this mistake of not removing this content provided in the template.

Line 140-149. Good work here!

Response: Thank you.

Line 147. “while” > “while SYM+I+G was selected for”

Response: Thank you.

Line 158 and 274. The word “basal” cannot be used like this. See https://resjournals.onlinelibrary.wiley.com/doi/10.1111/j.0307-6970.2004.00262.x - please use the “sister species” construction instead.

Response: Thank you. We accepted your suggestion Change “basal” to “a sister group”.

Line 158. “with a sound” > “with reasonable”

Response: Thank you.

Line 165. The “shortening of branches” has to do with the compromised multiple sequence alignment.

Response: Thanks for your suggestion. This issue has been resolved by rebuilding the phylogenetic tree and no shortening of branches

Line 287. “complied” > “compiled”

Response: Thank you.

Line 283. The key is a strong asset of this study. Nice work!

Response: Thank you.

Line 386. “Since” > “Due to”

Response: Thank you.

Line 393. “birth” > “inception”

Response: Thank you.

Line 393. “species have” > “delimitation has” or “circumscription has”

Response: Thanks for your suggestion. “species have” has been replaced with “delimitation has”

Line 394. I propose: “and currently, 47 species are accepted in the genus.”

Response: Thanks for your suggestion.

Line 395. “in classification” > “in the classification”

Response: Thank you.

Line 396. “explanations” > “explanation”. I think the authors should send an email to CoL and ask them to fix this issue. I’m sure the reason is that they are unaware of this complication.

Response: Thanks for your suggestion. We have sent an email to CoL to raise this issue with them.

Line 400. “good” > “strong”

Response: Thank you.

Line 476. “Fungi in Agricultural Soils” > “Fungi in agricultural soils”. Similarly on line 528.

Response: Thank you.

Round 2

Reviewer 2 Report

 I am, by and large, happy with the changes implemented by the authors. Some minor issues remain.

Line 13. ”sequences” > ”markers”

18. “feature” > “features”

18. “illustration” > “illustrations”

52. “as pathogen of the leaf blight of Poa pratensis” – I read this to mean that it is a pathogen of the [fungal] pathogen (leaf blight) of Poa pratensis. Is this the intended reading? If not, please clarify.

77. “at a” > “in a” ?

77. “3–4 days until” > “3-4 days, after which”

88. HSAUP is not recognized by Index Herbariorum (http://sweetgum.nybg.org/science/ih/).

105. “PCR programs” > “The PCR program”

Table:

Agrostis sp” . The “sp” should be “sp.” and not in italics.

“M sinense” > “M. sinense”

122. “locus” > “marker”

129. “, four” > “, and four”

131. “met, “ > “met “

148. “four-locus” > “four-marker”

155, 196, and 272. “supports” > “support values”

156. “is located” > “forms the”

159. “gene” can be deleted.

162. Is it still the case that “Some branches are shortened to fit to the page–these” , or was this an effect of the problematic sequences in the first submission round?

199. “are clustered together” > “form sister clades”. The authors did not use a clustering algorithm such as neighbour-joining and thus should not talk about “clustering”.

220. “sinense” is given italics here, but “miscanthi” is not given in italics on line 172. This strikes me as inconsistent.

235. “℃” seems to be written in a different font.

240. What is meant by “Details refer to the notes for M. miscanthi.” ? “For details, refer to the notes for M. miscanthi.” ?

273. “genes” > “genetic markers”

275. “a good” > “satisfactory”

276. Please clarify “conidiomata sporodochia”.

400. “four gene combinations” > “based on four genetic markers”

406. The authors use “Poaceae” in the table but “Graminae” here. This seems inconsistent. Or is a difference intended?

413. “four gene regions” > “the four genetic markers”

415. “Although a single gene region of” > “Although any of the genetic markers”

421. “sequences” > “multiple sequence alignment”.

Author Response

I am, by and large, happy with the changes implemented by the authors. Some minor issues remain.

Response: Thanks for your suggestion. Revisions have been highlighted.

Line 13. “sequences” > “markers”

Response: Thank you.

Line 18. “feature” > “features”, “illustration” > “illustrations”

Response: Thank you.

Line 52. “as pathogen of the leaf blight of Poa pratensis” – I read this to mean that it is a pathogen of the [fungal] pathogen (leaf blight) of Poa pratensis. Is this the intended reading? If not, please clarify.

Response: Thanks for your suggestion. We have deleted “of the leaf blight” to avoid semantic confusion.

Line 77. “at a” > “in a”?

Response: Thank you.

Line 77. “3–4 days until” > “3-4 days, after which”

Response: Thank you.

Line 88. HSAUP is not recognized by Index Herbariorum (http://sweetgum.nybg.org/science/ih/).

Response: Thanks for your suggestion. We have deposited holotype in the Herbarium Mycologicum Academiae Sinicae, Institute of Microbiology, Chinese Academy of Sciences, Beijing, China (HMAS) and isotype in Herbarium of the Department of Plant Pathology, Shandong Agricultural University, Taian, China (HSAUP).

Line 105. “PCR programs” > “The PCR program”

Response: Thank you.

Table:

“Agrostis sp” . The “sp” should be “sp.” and not in italics.

M sinense” > “M. sinense

Response: Thank you.

Line 122. “locus” > “marker”

Response: Thank you.

Line 129. “four” > “and four”

Response: Thank you.

Line 131. “met,” > “met “

Response: Thank you.

Line 148. “four-locus” > “four-marker”

Response: Thank you.

Line 155, 196 and 272. “supports” > “support values”

Response: Thank you.

Line 156. “is located” > “forms the”

Response: Thank you.

Line 159. “gene” can be deleted.

Response: Thanks for your suggestion.

Line 162. Is it still the case that “Some branches are shortened to fit to the page–these”, or was this an effect of the problematic sequences in the first submission round?

Response: Thanks for your suggestion. Sorry for the omission here. It has now been removed.

Line 199. “are clustered together” > “form sister clades”. The authors did not use a clustering algorithm such as neighbour-joining and thus should not talk about “clustering”.

Response: Thanks for your suggestion. We accepted your suggestion.

Line 220. “sinense” is given italics here, but “miscanthi” is not given in italics on line 172. This strikes me as inconsistent.

Response: Thanks for your suggestion. We have italicized “miscanthi”.

Line 235. “℃” seems to be written in a different font.

Response: Thanks for your suggestion. It has been adjusted to the same font.

  1. What is meant by “Details refer to the notes for M. miscanthi.”? “For details, refer to the notes for M. miscanthi.”?

Response: Thanks for your suggestion. We have revised as “For details, refer to the notes for M. miscanthi”.

Line 273. “genes” > “genetic markers”

Response: Thank you.

Line 275. “a good” > “satisfactory”

Response: Thank you.

Line 276. Please clarify “conidiomata sporodochia”.

Response: Thanks for your suggestion. Conidiomata and sporodochia are almost the same character, and we decid to keep sporodochia in the revised ms.

Line 400. “four gene combinations” > “based on four genetic markers”

Response: Thank you.

Line 406. The authors use “Poaceae” in the table but “Graminae” here. This seems inconsistent. Or is a difference intended?

Response: Thanks for your suggestion. The Graminae has been replaced by Poaceae.

Line 413. “four gene regions” > “the four genetic markers”

Response: Thank you.

Line 415. “Although a single gene region of” > “Although any of the genetic markers”

Response: Thank you.

Line 421. “sequences” > “multiple sequence alignment”.

Response: Thank you.